# Thai SMEs' Response in the Digital Economy Age: A Case Study of Community-Based Tourism Policy Implementation

**Wannapa Tongdaeng \***  **and Chandra-nuj Mahakanjana**

Graduate School of Public Administration, National Institute of Development Administration, Bangkok 10240, Thailand; chandra-nuj@nida.ac.th
\* Correspondence: wannapa.tongd@stu.nida.ac.th

**Abstract:** The goals of this study were to identify factors affecting the responses of small- and medium-sized enterprises (SMEs) in the digital economy age, to examine the impact of policy implementation and stakeholders' roles in the promotion of SMEs in the service sector of community-based tourism (CBT), to analyze the competitiveness of CBT, and to reveal the gap in SMEs' service quality. The research design consisted of both qualitative and quantitative methods. It was found that independent variables, i.e., the strategic management, the decision-making process, the structural contingency, perception, and the SMEs' potential, together in the multiple regression model, could predict dependent variables. Policy actors can be divided into three levels. Politicians are at the national policy level, high-level bureaucrats are at the implementation level, and stakeholders in the community are at the local level. Policy instruments include projects to support SMEs. Local-level actors play a role in promoting SMEs through collaborative public management. The competitiveness analysis of CBT can be divided into five forces: the rivalry of CBT, the threat of new entrants, the threat of substitutes, the bargaining power of suppliers, and the bargaining power of customers. Meanwhile, there are some problems in assuring service quality.

**Keywords:** SMEs' response; digital economy; community-based tourism; policy implementation; collaborative public management; service quality



## 1. Introduction

Because of the fourth digital revolution, each country has witnessed the transition to the era of the digital economy, and the form of economic activities has been changing immensely. The development of a digital system for such economic activities will not only increase production efficiency and create market opportunities, but will also add to the global value chains (ASEAN Secretariat 2018). In Thailand, the government declared a national development policy based on the digital economy concept in 2015 through the implementation of the Thailand 4.0 policy. This policy aims to develop the national economy through information technology applications in economic activities (Kiatsanpipop 2015), especially by boosting the digital economy through small- and medium-sized enterprises (SMEs) in the tourism sector (The Office of SMEs Promotion 2017), which affect the total economic growth of the country.

The Thai government aims to directly distribute income from tourism to people at the local level in the form of community-based tourism (CBT). CBT not only answers the needs of tourists but also helps to increase the capacity of each community without destroying the existing ways of life with the six following approaches: (1) improving the environmental friendliness and quality of tourism, (2) rebranding the image of Thai tourism in the eyes of global tourists, (3) using technologies to promote tourism in accordance with the concept of Thailand 4.0, (4) prioritizing the use of social media for the public relations of the country's tourism industry, (5) improving services and facilities according to the international standard, and (6) distributing tourism income to people at the local level

(Ministry of Tourism and Sports 2017). However, in many areas, SMEs in the tourism sector are facing the problem of adjustment due to digital technology changes, especially with the increase in the complexity of digital marketing (Krungthai Compass 2019). In addition, a survey on Thai SMEs during the past five years by TMB Analytics found that 38% of Thai SMEs are not yet ready for new technologies for at least three reasons. First, they are worried that the new startup will be troublesome. Second, they have no time for research and innovation. Finally, they deem that the existing actions are already good enough (Techsauce 2018). For this reason, it is obvious that the adjustment of Thai SMEs to cope with a swiftly changing environment in the digital economy would highly depend on the SME entrepreneurs' perception of opportunity and threat within the development stream of the digital economy. In other words, the entrepreneurs' perceptions can affect their decisions and actions to ensure that their own SMEs can respond well to the changing environment, in terms of both opportunity and threat. This corresponds to the structural contingency theory developed by Donaldson (2001), according to which an organization management team adjusts its organizational structure in accordance with the environment in order to ensure efficiency, effectiveness, and survival. Hatch and Cunliffe (2006) further explained that as the organization environment is complex, especially regarding technological development, which is constantly changing, the organization must attempt to respond to its environment mainly through a proactive strategy. Additionally, the organization members must be creative and focus on the strategy's effectiveness in reaching the goal. Currently, the actions of Thai SMEs still require much adjustment to cope with the major technological changes in the digital economy age.

The background of the fourth industrial revolution and Thai SMEs' business actions within the development stream of the digital economy as described above led to this study, entitled "Thai SMEs' response in the digital economy age: A case study of community-based tourism policy implementation". The upper northern provinces, which consist of four minor cities—Chiang Rai, Nan, Phayao, and Phrae—were selected as the focus of the study, due to their status as the third most popular destination for tourists, followed by the southern and eastern regions, which are the seaside attractions. In 2018, before the coronavirus disease 2019 pandemic (COVID-19), the total number of tourists who visited the northern region reached 24,163,245, representing a 4.42% growth compared to the previous year (Ministry of Tourism and Sports 2018). The purposes of this study were set to investigate SMEs' responses in the digital economy age and to examine three related factors—the context of CBT policy implementation and stakeholders' roles in SMEs' promotion within the framework of digital economic development, the competitiveness of SMEs in CBT attractions, and the gap in SMEs' service quality—to provide policy recommendations to help SMEs to adapt, survive, and grow sustainably. The research period was from 1 March 2019 to 31 December 2020.

## 2. Literature Review

Based on the review of the literature related to this study, the following theories and related research are described in this section.

### 2.1. Structural Contingency Theory

The basic concept of this theory is that no single type of organizational structure can always yield maximum efficiency for all organizations, as the appropriate structure depends more on the context of the organization (Donaldson 1999). Each type of organizational structure yields different results, and no structure is always the best (Galbraith 1977). Additionally, an organization that largely depends on its environment is definitely influenced by such an environment; its structure must therefore also be adjusted (Bowornwathana 2015). In general, the contingency theory can explain an open organization (Simon 1999) that must adapt itself in accordance with the environment, in terms of both size and the strategies of the organization (Donaldson 2001). Since the current external environment, especially

global competition, presents a challenge to the organization, executives are pressured to seek the best option for the organization to survive (Daft 2001).

### 2.2. The SMEs' Management Potential

Organization management resources are an important input factor required for a successful operation. Scholars such as Ferrero diRoccaferrera (1972), Rostamzadeh and Sofian (2011), Heizer and Render (2014), Pride et al. (2014), Anyaehie (2016), Wulandari and Rahmah (2020), and Kenaphoom (2014) have studied input factors in organization management during the last five decades. Most of them identified similar factors, namely man, money, management, material, machine, morality, market, message, method, minute, and mediation. If a large amount of quality management resources are available to an enterprise, this could reflect that such an enterprise has high capacity (Kenaphoom 2019). Therefore, a SMEs' operation, which aims to ensure the business's success within the constantly dynamic environment, requires an adequate number of reserved, quality resources.

### 2.3. Perception Theory

Generally, people can perceive attitudes, emotions, or conditions through interpretation by observing people's behaviors and the environment (Bem 1972). Similarly, organization executives must perceive and interpret the uncertain environment (Sutcliffe 1994) that may affect the organization. Executives' perceptions can be highly influential in ensuring that the organization can adjust itself to fit the changing environment (Yasai-Ardekani 1986). Additionally, these perceptions also significantly affect the organization's output (Waller et al. 1995), as the level of executives' awareness of the organization's environment is related to the level of priority in the strategic planning process (O'Regan et al. 2007). Especially in the age of technological change, executives are increasingly required to possess knowledge about technologies as well as their connection to technological development, which could reduce the risks of business (Tyler and Steensma 1998).

### 2.4. Decision-Making Theory

This theory focuses on the process and activities related to decisions and includes three sub-theories, as follows (Anderson 2014). Firstly, the rational-comprehensive theory aims to explain the decision process through a rationale that concerns the top goal of an organization. Secondly, the incremental theory aims to explain the decision-making process that can respond to the actual situation better than the rational-comprehensive theory. Thirdly, the mixed scanning theory originated from the belief that both previous theories have some weaknesses. This theory combines the strengths of the rational-comprehensive theory and of the incremental theory. It can be said that executives tend to be rational persons and therefore that they can manage their organizations to fit the external environment to survive (Lorsuwannarat 2015) based on the process of decision making, which depends on the executive's perception of both positive and negative consequences for the organization, social expectations towards the organization, and the ability to control internal resources to overcome obstacles (Harrison et al. 1997).

### 2.5. Strategic Management Theory

Currently, the organizational environment is complex and dynamic, and thus the executive must adjust the strategy, structure, and management in response to new business opportunities (Miles et al. 1995). Additionally, due to increased marketing competition, the executive must be committed to developing innovation to improve goods and services. Quality management is considered a potential internal strategy that can lead to innovation in the working process and product, as efficient strategic management has a positive relationship with organizational innovation (Schniederjans and Schniederjans 2015). Moreover, in the entrepreneurial ecosystem and the trend of technological changes, the most suitable strategy is a proactive one, in which digital entrepreneurs seek new opportunities and markets (Hatch and Cunliffe 2006; Salamzadeh and Ramadani 2021).

### 2.6. Five Forces Model

This model has been highly influential in business (Grundy 2006), and it is also a powerful framework to understand the competition faced by many enterprises. The competition is controlled by the following forces (Porter 2004): (1) barriers to entry, e.g., the economy of scale, product differentiation, a large amount of capital required for investment, switching costs for changing to a new product, access to distribution channels, and cost disadvantages independent of scale; (2) the intensity of rivalry among existing competitors, e.g., equally balanced competitors, slow industry growth, high fixed storage costs, lack of differentiation of switch costs, diverse competitors in the industry, high strategic stakes, and high exit barriers; (3) the pressure from substitutes, occurring when a new product emerges to replace the existing one; (4) the bargaining power of buyers, occurring when customers can bargain for a lower price while demanding a higher quality of services; and (5) the bargaining power of suppliers, occurring when suppliers can either increase the raw material cost or decrease the quality of raw materials.

### 2.7. The Gap Model of Service Quality

Service quality refers to the ability of an organization to meet the customer's expectations. The customers wish to receive a service that is as expected (Parasuraman et al. 1994). Accordingly, service businesses require gap model analysis to learn about their service quality by measuring satisfaction at each step. Through this, businesses can improve efficiency to reach a higher standard in the future (Rosene 2003). The elements of service quality can be divided into three dimensions: physical facilities, operating process, and service staff's behavior (Haywood-Farmer 1988). Additionally, the general method of evaluating service quality is usually the evaluation of the gap between the customers' expectations and their perception of the actual service. In the gap model, service quality is measured by "RATER", which utilizes five factors: reliability, assurance, tangibles, empathy, and responsiveness (Marathe 2017).

## 3. Research Method

### 3.1. Research Design

This research consisted of a mixed methods approach to meet the four objectives of the study: to uncover factors affecting the SMEs' response in the digital economy age, to examine the context of policy implementation and the stakeholders' roles in the promotion and development of SMEs of CBT within the framework of digital economy development, to analyze the competitiveness of SMEs, and to point out the gap in the service quality of SMEs.

### 3.2. Variables

The dependent variable was the SMEs' response in the digital economy age. The SMEs' response referred to the operational guidelines, which reflected the perception and the decision-making process of entrepreneurs during CBT policy implementation. The independent variables involved the five following factors (Table 1).

**Table 1.** Independent variables in the study.

| Independent Variables | Authors of Related Research |
|---|---|
| 1. The structural contingency factors refer to the factors of the open organization structure. Entrepreneurs used this organizational structure to allow for flexibility, the decentralization of authority, teamwork, low formality with no emphasis on control, and informal internal communication. | Ellis et al. (2002); Schoonhoven (1981); Chenhall (2003); Khazanchi (2005); Sun and Pan (2011) |
| 2. The SMEs' potential factor refers to the factors that reflect that SMEs had enough management resources, considering the operation's inputs, i.e., money, men, materials, methods, machines, and marketing. | Phisarnchananan et al. (2018); Wulandari and Rahmah (2020) |
| 3. The perception factors refer to the entrepreneurs' perception of e-business infrastructure, which supports the electronic business process; e-business, which is the process by which the business sector operates on the computer network; and e-commerce, which is the trading of goods and services on the computer network. | Hornsby et al. (2002); Lin and Lee (2004); Hornsby et al. (2009); Thomas and Lamm (2012); Khuong and An (2016) |
| 4. The decision-making process factors refer to the factors related to the entrepreneurs' decision-making process in digital leadership in choosing or avoiding the implementation of technology, such as the positive and negative effects on enterprises, social expectations, and the ability to control internal resources. | Rowe et al. (1984); Ivancevich et al. (1990); De Clercq et al. (2011); Harrison et al. (1997); Shepherd (2011); Salamzadeh et al. (2021) |
| 5. The strategic management factors refer to strategies chosen by entrepreneurs to gain opportunities and handle threats from technology to achieve the business outcome. The strategies can be classified into four types: the Prospectors strategy, the Defenders strategy, the Analyzers strategy, and the Reactors strategy. | Singh et al. (2008); Kraus and Kauranen (2009); Pertusa-Ortega et al. (2010) |

*3.3. Conceptual Framework*

Based on the consideration of the literature review of SMEs' importance in the digital economy age, the conceptual framework of this study is divided into four parts, as follows (Figure 1).

Part I is the study of factors influencing the SMEs' response, which reflects the perception and decision-making process of entrepreneurs during CBT policy implementation to ensure their adaptation, survival, and growth during the era of the digital economy.

Part II is the study of policy implementation, SMEs' adjustment, and stakeholders' roles in the promotion and development of CBT under the framework of digital economy development to understand the operational mechanisms, the parties concerned, and the tools for implementing CBT policy to reach the goals.

Part III is the study of the competitiveness analysis of SMEs in CBT attractions to reflect the potential, contexts, and adjustments of SMEs that are supported by government policy. The Five Forces model is applied in this part.

Part IV is the study of the gap in the service quality of SMEs' operations to raise awareness of the problems faced by SMEs and make suggestions to enable SMEs to adapt, survive, and grow. The service quality was measured by the satisfaction of tourists who used the tourism-related services from SMEs. The scope of satisfaction measurement included reliability, assurance, tangibles, empathy, and responsiveness, also known as "RATER" (Marathe 2017).

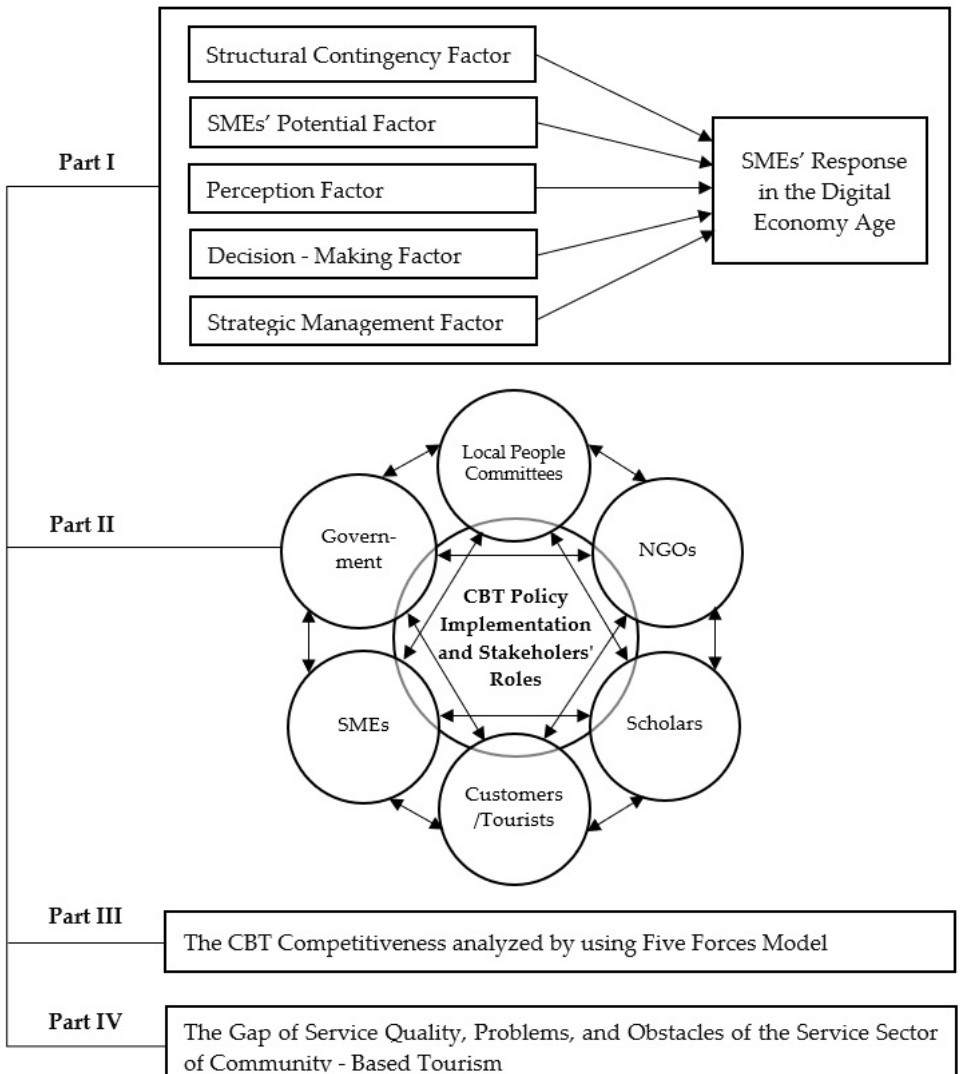

**Figure 1.** Conceptual framework of the study.

*3.4. Hypothesis*

The hypothesis was formulated from the conceptual framework of Part 1.

**Hypothesis 1 (H1).** *All independent variables in the multiple regression equation cannot together affect the response of SMEs in the digital economy age (H1: $\rho = 0$).*

**Hypothesis 2 (H2).** *All independent variables in the multiple regression equation can together affect the response of SMEs in the digital economy age (H2: $\rho \neq 0$).*

The independent variables included five factors related to SMEs' operations, all of which were based on the review of the literature in this study. They were (1) the structural contingency, (2) the SMEs' potential, (3) the perception, (4) the decision-making process, and (5) the strategic management. The dependent variable was the SMEs' responses to opportunities and threats in the digital economy age.

*3.5. Population and Sampling*

The population included six groups of stakeholders in the promotion and development of CBT. The areas were selected from the upper northern provinces group 2, namely, Chiang Rai, Nan, Phayao, and Phrae. A total of 16 attractions based on the survey by the Office of the Permanent Secretary, the Ministry of Tourism and Sports, were identified to enact

an action plan to promote sustainable and creative CBT in the years 2018–2022. Only one of the CBT attractions per province was selected as a target area. The purposive sampling process for data collection was used to select the samples from the business sector, related government agencies, local people's committees, scholars, and non-governmental organizations (NGOs). Additionally, accidental sampling was used to select tourists from four CBT attractions. The sample size and instruments for the data collection of each group are shown in Table 2.

**Table 2.** Sample groups and instruments for data collection.

| Sample Groups | Number | Instruments * |
|---|---|---|
| 1. Business Sector | | |
| 1.1 SMEs' entrepreneurs | | |
| - At least 40 entrepreneurs (80.0% of a number) from each community for the survey; each community had approximately 40–50 service businesses. | 160 ** | Questionnaire |
| - Three entrepreneurs from each community to form a focus group | 12 | Interview guide |
| 1.2 Representatives of the Provincial Federation of Thai Industries, with one representative from each community for the in-depth interview | 4 | Interview guide |
| 1.3 Representatives from the Provincial Chamber of Commerce, with one representative from each community for the in-depth interview | 4 | |
| 2. Related government agencies for the in-depth interview | | |
| 2.1 Provincial Tourism and Sports Office's representatives | 4 | |
| 2.2 Tourism Authority of Thailand's representatives from three branches: Nan, Phrae, and Chiang Rai—Phayao | 3 | Interview guide |
| 2.3 A representative of the designated areas for sustainable tourism administration (DASTA) | 1 | |
| 2.4 Local administrative organization's representatives from four communities, with one representative from each community | 4 | |
| 3. Representatives of local people's committees from four communities, with three representatives from each community, for the in-depth interview | 12 | Interview guide |
| 4. Representative scholars of the local-educational institutes, one representative from each community, for the in-depth interview | 4 | Interview guide |
| 5. Representatives of NGOs in four communities, with one representative from each community, for the in-depth interview | 4 | |
| 6. Representatives of tourists who visited the CBT attractions | | |
| - Three representatives of tourists from each community for the focus group | 12 | Interview guide |
| - 100 representatives of tourists from each community for the survey | 400 *** | Questionnaire |
| Total | 624 | |

Note: * All instruments received a certificate of approval from the Ethics Committee in Human Research, National Institute of Development Administration, with COA number: 2020/0023. ** Calculated by G*Power, version 3.1.9.4 (The G*Power Team 2017). *** Calculated by Cochran's formula for infinite population (Cochran 1977).

*3.6. Data Analysis*

The data analysis was divided into two parts: Part 1 concerned the qualitative analysis. Content analysis was used to reveal the context of policy implementation and stakeholders' roles in terms of the promotion and development of SMEs, along with competitiveness, i.e., destinations, the demand for goods and services, related suppliers, the CBT strategic plan, and governmental support. Part 2 dealt with the quantitative analysis, including (1) descriptive statistics: the frequency, percentage, arithmetic mean, and standard deviation, to reveal the service quality gap of SMEs; and (2) inferential statistics to identify factors influencing the SMEs' responses. Multiple regression analysis was used to examine the

relationship between the dependent variable (SMEs' Res) and each of the independent variables $X_1$, $X_2$, ... , which needed to be a linear relationship.

The equation for the multiple regression analysis with unstandardized coefficients:

$$Y_{SMEs' Res'} = a + b_1X_1 + b_2X_2 + \ldots + b_nX_n$$

The equation for the multiple regression analysis with standardized coefficients:

$$Y_{Z' SMEs' Res} = \beta_1 X_{z1} + \beta_2 X_{z2} + \ldots + \beta_n X_{zn}$$

It should be noted that, prior to the analysis of the influence of independent variables on the dependent variable, it was necessary to uncover the *r* value (between 0.00 and 1.00) among the independent variables to prevent the problem of multicollinearity. The *r* value between them was not too high or too close to 1.00 (Farrar and Glauber 1967).

## 4. Findings and Discussion

The analytical results were as follows: (1) for the SMEs' responses to opportunities and threats in the digital economy age, $\bar{x} = 4.26$; SD = 1.09. (2) For the structural contingency factor (Struc), $\bar{x} = 4.33$; SD = 1.12. (3) For the SMEs' potential factor (Poten), $\bar{x} = 4.39$; SD = 1.08. (4) For the perception factor (Per), $\bar{x} = 4.43$; SD = 1.19. (5) For the decision-making process factor (Decis), $\bar{x} = 4.41$; SD = 1.16. (6) For the strategic management factor (Stra), $\bar{x} = 4.25$; SD = 1.15. The coefficient of variation (CV) of each variable was 0.26, 0.26, 0.25, 0.27, 0.26, and 0.27, respectively. All the predictor variables fulfilled statistical conditions; there was no collinearity problem between these variables. The statistical conditions were that (1) the level of tolerance must be higher than 0.19; the formula to calculate the level of tolerance is $1—R^2$, where $R^2$ is the amount of variance within the variable $X_1$ that can be explained by other predictor variables, namely, $X_2$, $X_3$, ... , $X_n$ within the equation); and (2) the variance inflation ratio (VIF) must be lower than 5.3 (Hair et al. 2010) (Table 3).

**Table 3.** Mean, standard deviations, and coefficient correlations.

|  | Mean | SD | CV | SMEs' Res | Struc | Poten | Per | Decis |
|---|---|---|---|---|---|---|---|---|
| SMEs' Res | 4.26 | 1.09 | 0.26 |  |  |  |  |  |
| Struc | 4.33 | 1.12 | 0.26 | 0.713 ** |  |  |  |  |
| Poten | 4.39 | 1.08 | 0.25 | 0.739 ** | 0.762 ** |  |  |  |
| Per | 4.43 | 1.19 | 0.27 | 0.810 ** | 0.670 ** | 0.703 ** |  |  |
| Decis | 4.41 | 1.16 | 0.26 | 0.827 ** | 0.706 ** | 0.754 ** | 0.833 ** |  |
| Stra | 4.25 | 1.15 | 0.27 | 0.849 ** | 0.698 ** | 0.765 ** | 0.857 ** | 0.821 ** |

$n = 160$; ** $p < 0.01$, Tolerance > 0.19 (Struc = 0.37, Poten = 0.30, Per = 0.21, Decis = 0.23, Stra = 0.20); VIF < 5.30 (Struc = 2.70, Poten = 3.36, Per = 4.71, Decis = 4.31, Stra = 4.97).

Based on the results of the analysis of factors influencing the SMEs' response by using five steps of the multiple regression analysis, the coefficients of multiple determinations ($R^2$) of Models 1–5 were found to be 50.8%, 60.0%, 72.7%, 75.7%, and 78.7%, respectively. Regarding the size of the influence of all of the five predictor variables over the SMEs' response (SMEs' Res), it was found that the variable Struc had a low level of positive influence ($\beta = 0.122$), the variable Poten had a very low level of positive influence ($\beta = 0.050$), the variable Per had a low level of positive influence ($\beta = 0.133$), the variable Decis had a moderate level of positive influence ($\beta = 0.277$), and the variable Stra had a moderate level of positive influence ($\beta = 0.383$) (Table 4).

**Table 4.** Multiple regression analysis.

| Variables | Model 1 | | | | Model 2 | | | | Model 3 | | | | Model 4 | | | | Model 5 | | | |
|---|---|---|---|---|---|---|---|---|---|---|---|---|---|---|---|---|---|---|---|---|
| | b | beta | t | *p* | b | beta | t | *p* | b | beta | t | *p* | b | beta | t | *p* | b | beta | t | *p* |
| (Constant) | 1.262 | | 5.214 | 0.000 | 0.696 | | 2.918 | 0.004 | 0.343 | | 1.696 | 0.092 | 0.278 | | 1.448 | 0.150 | 0.291 | | 1.612 | 0.109 |
| Struc | 0.692 | 0.713 | 12.783 | 0.000 | 0.347 | 0.357 | 4.578 | 0.000 | 0.184 | 0.189 | 2.801 | 0.006 | 0.138 | 0.142 | 2.197 | 0.030 | 0.119 | 0.122 | 1.997 | 0.048 |
| Poten | | | | | 0.469 | 0.467 | 5.980 | 0.000 | 0.228 | 0.226 | 3.204 | 0.002 | 0.136 | 0.136 | 1.942 | 0.054 | 0.050 | 0.050 | 0.734 | 0.464 |
| Per | | | | | | | | | 0.479 | 0.524 | 8.513 | 0.000 | 0.296 | 0.324 | 4.399 | 0.000 | 0.122 | 0.133 | 1.651 | 0.101 |
| Decis | | | | | | | | | | | | | 0.333 | 0.354 | 4.403 | 0.000 | 0.261 | 0.277 | 3.587 | 0.000 |
| Stra | | | | | | | | | | | | | | | | | 0.362 | 0.383 | 4.616 | 0.000 |
| n | 160 | | | | 160 | | | | 160 | | | | 160 | | | | 160 | | | |
| R | 0.713 | | | | 0.774 | | | | 0.852 | | | | 0.870 | | | | 0.887 | | | |
| $R^2$ | 0.508 | | | | 0.600 | | | | 0.727 | | | | 0.757 | | | | 0.787 | | | |
| Adj.$R^2$ | 0.505 | | | | 0.595 | | | | 0.721 | | | | 0.751 | | | | 0.780 | | | |
| F | 163.40; df = 1, 158; $p = 0.000$ | | | | 117.56; df = 2, 157; $p = 0.000$ | | | | 138.21; df = 3, 156; $p = 0.000$ | | | | 120.72; df = 4, 155; $p = 0.000$ | | | | 113.49; df = 5, 154; $p = 0.000$ | | | |
| $\Delta R^2$ | 0.508 | | | | 0.092 | | | | 0.127 | | | | 0.030 | | | | 0.030 | | | |
| $\Delta F$ | 163.40; df = 1, 158; $p = 0.000$ | | | | 35.76; df = 1, 157; $p = 0.000$ | | | | 72.47; df = 1, 156; $p = 0.000$ | | | | 19.38; df = 1, 155; $p = 0.000$ | | | | 21.31; df = 1154; $p = 0.000$ | | | |

Therefore, the results of the analysis confirm the research hypothesis of this study, since all the independent variables in the equation could predict the dependent variable. The unstandardized and standardized coefficients in the fifth model of the multiple regression analysis were as follows.

$$\text{SMEs' Res'} = 0.291 + 0.119(\text{Struc}) + 0.050(\text{Poten}) + 0.122(\text{Per}) + 0.261(\text{Decis}) + 0.362(\text{Stra})$$

$$Z_{\text{SMEs' Res}} = 0.122(Z_{\text{Struc}}) + 0.050(Z_{\text{Poten}}) + 0.133(Z_{\text{Per}}) + 0.277(Z_{\text{Decis}}) + 0.383(Z_{\text{Stra}})$$

The theoretical contributions of the factors which together influenced the SMEs' responses in the digital economy age can be summarized as follows.

The strategic management factor (Stra): This factor influenced the response of SMEs at the highest level (38.3%). The finding confirmed the research hypothesis. According to strategic management theory, all enterprises must adjust their strategy and management process in response to the changing environment and new business opportunities (Miles et al. 1995). Especially in the current business environment, which is highly competitive, entrepreneurs must give a higher priority to thinking, planning, and strategic management (Harindranath et al. 2008). Strategic management of the organization should correspond well to the environment. Therefore, it is necessary to conduct a thorough survey of the environment before formulating a clear strategy for implementation (Analoui and Karami 2002). The organization may focus more on an innovative strategy rather than an internal cost control strategy in order to respond to the highly competitive environment (Tang and Hull 2012).

The decision-making process factor (Decis): This factor influenced the response of SMEs at 27.7%. The findings confirm the hypothesis of the decision-making theory, as Anderson (2014) explains that decision making based on reasoning takes into consideration achieving the highest goal of an individual or organization, which consists of six elements: (1) identifying the problem clearly, (2) having a clear goal, value, or objective, (3) finding options to reach the goal, (4) analyzing the costs and benefits or the strengths and weaknesses of each alternative, (5) comparing the alternatives to find the best one, and (6) deciding which alternative will achieve the goal most effectively. Similarly, Lorsuwannarat (2015) explained that executives tend to be reasonable, and therefore that they play a major role in organizing the enterprise to suit the external environment in order to ensure survival and effective operation to achieve the organizational goal. The decision-making process involves three factors (Harrison et al. 1997), which are (1) an awareness of both negative and positive effects of the external environment on the enterprise; (2) the expectations of the society about the enterprise, especially when the enterprise grows, as the society's expectation will also grow; and (3) an ability to control organizational resources to solve problems and obstacles.

The structural contingency factor (Struc): This factor affected the response of SMEs at 12.2%. Although the size of this variable's influence was not high, it could still confirm the hypothesis of the structural contingency theory formulated by Donaldson (2001) that the enterprise will focus on adjustment in line with the environment—that is, by adjusting its structure to suit the factors in the environment. For example, Raymond (2005) proposed that an organization can organize its internal management system and implement advance manufacturing technology to reduce costs while increasing production efficiency, product flexibility, quality of work, and integration of work among departments. Whenever the enterprise faces an increasingly uncertain environment, it is increasingly required to organize its operation system to match the environment. Likewise, Sun and Pan (2011) stated that the organization must adjust its structure to suit the market change, which will benefit the organization's effectiveness.

The perception factor (Per): This factor could influence the response of SMEs at 13.3%. Although the size of influence of this factor was small, it could still confirm the hypothesis of the perception theory that an individual could perceive the attitude, emotion, and condition by interpreting the observed behavior or the environment (Bem 1972). Similarly,

an executive perceives and interprets an uncertain situation (Sutcliffe 1994), which can impact the enterprise. Therefore, the executive's perception of the external environment can greatly influence the structural adjustment and internal management of the enterprise in the constantly dynamic environment (Yasai-Ardekani 1986). What the executive knows and is interested in can significantly influence the enterprise's operation (Waller et al. 1995). Nevertheless, it should be noted that the findings in this study correspond to those in the study of McAdam and Reid (2001) indicating that the perception of SMEs entrepreneurs in terms of knowledge management and innovative development is still unclear, especially SMEs in developing countries that lack investment in the systems and mechanisms which support the enterprise's knowledge management, such as e-commerce and e-business. It is due to the perspectives of entrepreneurs that their business does not gain much profit from investment and that the outcome of investment is low in comparison to investment by large-scale enterprises (MacGregor and Kartiwi 2010).

The SMEs' potential factor (Poten): This factor influenced the response of SMEs at 5.0%. Although the size of influence of this factor may be very low, it can still confirm the concept of 8Ms proposed by Ferrero diRoccaferrera (1972). He stated that the elements for organizational management competency are man, money, material, management, machines, market, message, and morale. It is necessary for the enterprise to prepare all resources. According to the study entitled "Competitive strategy, structure and firm performance: A comparison of the resource-based view and the contingency approach" conducted by Pertusa-Ortega et al. (2010), the enterprise needs to adjust the structure of the organization to suit the environment at that time, as restructuring the organization can lead to the efficiency of the operation through competitive strategy. Hence, it is not surprising that the MRA analysis in this study revealed that the SMEs' potential factor (Poten) was significantly related to the perception factor (Per), the decision-making process factor (Decis), and the strategic management (Stra).

Although the five independent variables could together influence the level of responses by SMEs to opportunities and threats in the digital economy age, it is more important to find a way to ensure that SMEs, which are the primary economic backbone of the country, can survive and grow securely under both internal and external pressures, such as technological changes, conditions to access loans from financial institutions, fiercely competitive marketing, natural disasters, and the COVID-19 pandemic. Therefore, the researchers studied and analyzed the context of the government's policy implementation in the promotion and development of SMEs to understand the policy, mechanisms, actors, stakeholders' role, instruments, and approaches so that SMEs can efficiently respond to economic opportunities and technological threats.

The operational mechanism was divided into three levels: national policy, implementation, and local (province and community) levels. When the researchers analyzed the mechanism based on the theory of policy implementation and reviewed the studies conducted by scholars such as Dellar (1992), Madon et al. (2007), and Johnson (2013), it was found that the three levels of the mechanism could be divided into two levels: (1) the macro level, which involved the mechanism at the national policy level, and the implementation level. The macro-level implementation consisted of two steps. Firstly, the policy had to be transformed into plans, projects, and guidelines (Rabinovitz et al. 1976). Secondly, the agencies at the implementation level had to be convinced to accept and be willing to implement the guidelines, plans, and projects (Hill and Hupe 2009; Lipsky 2010). Additionally, (2) the micro level involved the mechanism at the local level and consisted of three steps. The first step was "kickstarting". The local agencies performed two important activities: considering whether to accept the policy and seeking support from network partners (Lambright et al. 1977). The second step was "implementation". This step involved the process of adjusting plans and projects taken from the central administration and materializing them. Here, implementers could either change their behavior to fit plans and projects or adjust plans and projects to suit the characteristics of their regular tasks. The implementers could use their personal judgement in working together with the target groups, which the political

sector or the higher-level agencies could not oversee (Berman and McLaughlin 1974; Larsen and Agarwala-Rogers 1977; Majone and Wildavsky 1978). Finally, "ensuring cohesion and continuity" involved finding a way for the policy to be accepted and adjusted to fit the regular and continuous mission of the implementers (Meyers and Vorsanger 2007) (Figure 2).

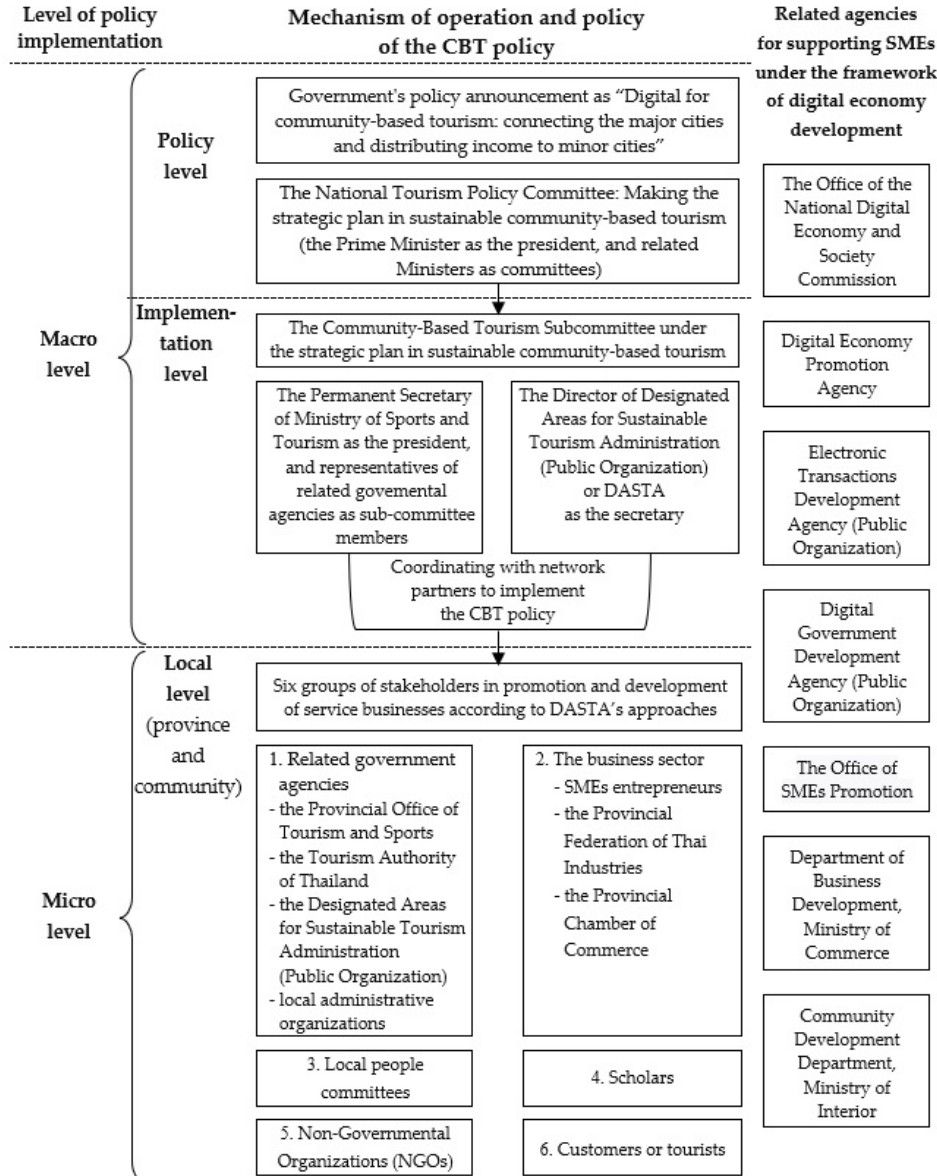

**Figure 2.** Level of policy implementation, mechanism of operation, actors, and related agencies for supporting SMEs.

The policy actors were those related to the policy's implementation to promote and develop service businesses in the CBT group under the framework of digital economy adjustment. These actors differed in expectations and goals but were required to interact or collaborate. The role and influence of each group over the outcome of policy implementation also differed (Bressers 2004), with no individual partner being able to control all of the results or the direction of policy implementation (Chandarasorn 2002). Hence, the actors could be divided into three levels: (1) the national policy level, the political sector, or the Cabinet. This level encompasses the prime minister and related ministers responsible for policy implementation. The political sector had a role in initiating and proposing the CBT policy under the concept "digital for CBT: connecting the major cities

and distributing income to minor cities" as well as in establishing rules and regulations and allocating budgets for policy implementation. (2) The implementation level refers to high-level bureaucrats belonging to the agencies mainly responsible for policy implementation and appointed by the National Tourism Policy Committee. (3) Additionally, the local level refers to the stakeholders in CBT development, especially in the province and the community. This group of actors had an important role in the success and failure of policy implementation, especially the government agencies that served as the government's operation mechanism, such as the Provincial Office of Tourism and Sports, the Tourism Authority of Thailand, and the local administrative organization. In addition, there were other supporting agencies, such as the Office of the National Digital Economy and Society Commission, the Digital Economy Promotion Agency, the Electronic Transactions Development Agency, the Digital Government Development Agency, the Office of SMEs Promotion, the Department of Business Development, and the Department of Community Development. Instruments and approaches consisted of plans and projects to develop service businesses within the framework of digital economy development formulated by the local government agencies, resulting from the transformation of the national policy into implementation (Howlett 1991; Lowi 1972) under the annual action plan of the agency. The approaches consisted of (1) delegating legal authority to the local agencies along with supporting budgets to carry out the action plan, (2) following the plans set prior to the project overseen by the executives and the committee to ensure the progress and success of the project, and (3) promoting a sustainable CBT development process from planning, action, and evaluation to improvement.

As for the role of the local-level actors in policy implementation, there were six groups of stakeholders: the business sector, related government agencies, local people's committees, scholars, NGOs, and tourists. All of them played a role in the promotion and development of service businesses in CBT in accordance with the concept of collaborative public management (CPM), which is part of collaborative governance (Emerson et al. 2012). The collaboration was horizontal among the diverse groups, organizations, and institutions to implement the same policy with a high flexibility and mutually dependent interaction among the actors, especially at the community level (Agranoff 2003; Bingham 2008; Kapucu et al. 2009). The researcher divided the goal of collaboration based on the idea of Huxham et al. (2009), who divided the goal in terms of the relevance and substance of collaboration. It was found that the collaborative partners all played a role related to the promotion and development of service businesses in CBT. The agency that was mainly responsible for the horizontal collaboration with related agencies at the local level was DASTA. The essence of collaboration was divided into four dimensions (Designated Areas for Sustainable Tourism Administration 2018): (1) developing human resources, the community, and the network to ensure the community's ability to link with others and mutually exchange resources, knowledge, and experiences as a strong network; (2) increasing the value of local resources by seeking the original resources of the community and linking them to tourism; (3) developing the marketing techniques of each community to serve the tourists' demand and encouraging entrepreneurs to arrange tourism activities together to add value to the tourism route; and (4) making sure that all sectors participate through the process of mutual knowledge exchange.

After the government announced the policy of promotion and development for CBT businesses in 2016, the CBT industry of Thailand grew with higher competency, as evidenced by the result of a competency analysis of CBT based on Porter's Five Forces model (Porter 2004). The findings can be itemized as follows. First, in the rivalry CBT, it was found that there were many emerging SMEs in CBT attractions, but when the supply in the industry became too large, the market became highly competitive. Therefore, SMEs with larger capital that could adjust their marketing strategy well had a better chance of surviving and growing. Regarding the threat of new entrants, it was found that for many CBT attractions, there were many emerging SMEs that provided similar products and services. Additionally, SMEs with higher bargaining power needed to have skills and

experience in service businesses, business alliances, unique products, a higher economy of scale, a low cost of switching for their customers, diverse distribution channels, and continuous support from the government sector. Regarding the threat of substitutes, it was found that there was an opportunity for more and more new tourist attractions to emerge, as communities tended to imitate and develop new attractions to compete with existing attractions. Moreover, the tourism-promotion policy of the government tended to continue with support for both existing and newly emerging tourist attractions. Regarding the bargaining power of suppliers, it was found that most bargaining power was in the form of mutual dependency, such as in tour agencies, agricultural producers, and labor in the service sector. Additionally, regarding the bargaining power of customers, it was found that most customers studied and compared the quality of products and services prior to their trip and thus had a rather high bargaining power. Meanwhile, tourism management by the community was not flexible, as prior contact with the community was required before every trip.

Although the CBT industry has been growing continuously during the past five years, it was found that customers' opinions, from 400 representatives from four communities, on the quality of SMEs' services still revealed a gap in quality, especially in terms of the readiness of the system and traffic infrastructure, closed-circuit cameras, security staff, signs with details of service conditions, and the adequacy and quality of facilities. Therefore, entrepreneurs should solve these problems as soon as possible (Lee et al. 2016) and focus on related factors, e.g., economic factors, social and cultural factors, and marketing (Salamzadeh et al. 2021), to increase customers' trust and satisfaction. However, although entrepreneurs had already improved the quality of service, this could not guarantee that customers would be willing to increase their payment for better-quality services (Ham et al. 2003; Zeithaml et al. 1996). Entrepreneurs should consider service quality improvement based on the return on investment first.

## 5. Conclusions and Recommendations

### 5.1. The Factors Affecting SMEs' Response in the Digital Economy Age

It was found that all the independent variables together in the multiple regression model could predict the dependent variables. The strategic management factor is the variable that confirms the assumption of strategic management theory. SMEs' entrepreneurs in the CBT need to adjust their strategies and management process to the changing environment in order to obtain new business opportunities (Miles et al. 1995), especially in the highly competitive business environment. Therefore, it is necessary for SMEs' entrepreneurs to give a higher priority to planning and strategic management (Harindranath et al. 2008). The decision-making process factor is the variable that confirms the assumption of the decision-making theory proposed by Anderson (2014) that a logical decision requires consideration of how to achieve the highest goal of an organization. Likewise, Lorsuwannarat (2015) explained that executives tend to make decisions based on reasoning, and they play a major role in managing an organization to suit the external environment to ensure the growth and effective operation of the organization. The structural contingency factor is the variable that confirms the assumption of the structural contingency theory. In this study, it was found that the majority of SMEs had structural flexibility and decentralized power among departments. The organizational management was in line with the structural contingency theory, which could be called organic organization or living organization, and Donaldson (2001) and Raymond (2005) explained that an organization will adjust itself to fit its environment by adjusting its structure to cope with the factors in the environment at the time. The perception factor is the variable that confirms the assumption of the perception theory. An individual can perceive attitudes, emotions, and conditions by interpreting the observed behavior or environment (Bem 1972; Sutcliffe 1994; Yasai-Ardekani 1986). Meanwhile, SMEs' entrepreneurs perceive and interpret the uncertain environment, which can affect their business. Therefore, the entrepreneurs' perception of the external environment can greatly influence structural reorganization and internal

administration to ensure adjustment and survival in the constantly dynamic environment. The SMEs' potential factor is the variable that confirms the assumption of the concept of 8Ms proposed by Ferrero diRoccaferrera (1972). He stated that each organization needs to have sufficient resources in both quality and quantity to reach the organization's goal of management in the ever-changing environment. In particular, in the digital economy age, SMEs' entrepreneurs need to apply new technologies, information, and online platforms to grow, increase their competitiveness, and support decision making in choosing the most appropriate strategy for their organizations.

### 5.2. The Context of Policy Implementation and the Role of Stakeholders

Operational mechanisms were divided into three levels: national policy, implementation, and local (province and community) levels. When the researchers analyzed these mechanisms based on the theory of policy implementation and reviewed the studies conducted by scholars such as Dellar (1992), Madon et al. (2007), and Johnson (2013), it was found that the three levels of mechanisms could actually be divided into two levels: the macro level, which involved the mechanisms at the national policy level and at the implementation level, and the micro level, which involved the mechanisms at the local level. The policy actors were those related to the implementation of policy to promote and develop service businesses in CBT groups within the framework of digital economy adjustment. These actors differed in terms of expectation and goals. However, they were required to interact or collaborate. The role and influence of each group over the outcome of policy implementation also differed (Bressers 2004), with no individual partner being able to control the whole result or the direction of policy implementation (Chandarasorn 2002). Hence, the actors could be divided into three levels: (1) the national policy level, or the political sector or the Cabinet, (2) the implementation level, referring to high-level bureaucrats of the agencies mainly responsible for policy implementation, and (3) the local level, referring to the stakeholders in CBT development. Instruments and approaches consisted of plans and projects to develop service businesses in CBT within the framework of digital economy development formulated by the local government agencies, which is also the result of transforming the national policy into implementation (Howlett 1991; Lowi 1972). The local stakeholders or the local-level actors in policy implementation played a role in the promotion and development of service businesses in CBT in accordance with the concept of collaborative public management (CPM), which is part of collaborative governance (Emerson et al. 2012). Horizontal collaboration occurs among the diverse groups, organizations, and institutions to implement the same policy with high flexibility and mutually dependent interaction among the actors, especially at the community level (Agranoff 2003; Bingham 2008; Kapucu et al. 2009). The researchers divided the goal of collaboration based on the idea of Huxham et al. (2009), who divided the goal in terms of the relevance and substance of collaboration. It was found that the collaborative partners all had a role related to the promotion and development of service businesses in CBT.

### 5.3. The Competitiveness Analysis of CBT According to Porter's Five Forces Model

As evidenced by the result of a competency analysis of CBT based on Porter's Five Forces model (Porter 2004), the findings may be itemized as follows: (1) rivalry of community-based tourism. It was found that there were many emerging CBT SMEs in the service sector, but when the supply in the industry became too great, the market became highly competitive. Therefore, SMEs with more capital who could adjust their marketing strategy well would have a better chance of surviving and growing. (2) Threat of new entrants. It was found that, in many CBT attractions, there were a large number of emerging SMEs that provided similar products and services, and SMEs with higher bargaining power have skills and experience in service businesses, business alliances, unique products, higher economies of scale, low costs of switching among their customers, diverse distribution channels, and access to continuous support from the government sector. (3) Threat of substitutes. It was found that there was an opportunity for more and more new tourist attractions to

emerge, as communities tended to imitate and develop new attractions to compete with the existing attractions. Moreover, the tourism-promotion policy of the government tended to continue with the support for both the existing and the newly emerging tourist attractions. (4) Bargaining power of suppliers. It was found that most bargaining power was in the form of mutual dependency, such as tour agencies, agricultural producers, and labor in the service sector. (5) Bargaining power of customers. It was found that most customers studied and compared the quality of products and services prior to their trip and thus had rather high bargaining power. Meanwhile, tourism management by the community was not flexible as prior contact with the community was required before every trip.

*5.4. The Gap in Service Quality of the CBT Businesses*

Most service businesses still had a gap in quality in terms of customer confidence. The survey results reveal that customers trust the service only at a moderate level, which was lower than their expectation, which was that they would desire it at a high level, especially in terms of the readiness of the system and traffic infrastructure, closed-circuit cameras, security staff, signs with details of service conditions, and the adequacy and quality of the facilities. Therefore, entrepreneurs should solve these problems as soon as possible while also regularly reviewing the readiness and adequacy of facilities (Lee et al. 2016) in order to increase customers' trust and satisfaction. However, although the entrepreneurs had already improved the quality of service, they could not guarantee that customers would be willing to increase their payment for better-quality services (Ham et al. 2003; Zeithaml et al. 1996).

*5.5. Policy Recommendations*

In accordance with the results of the study of problems, obstacles, and suggestions from stakeholders for the promotion and development of SMEs in the service sector of CBT, the policy recommendations were made in four dimensions as follows.

Firstly, digital technology skills: since SME entrepreneurs lack knowledge and skills to implement digital technology for business operations, the government should focus on enabling them to adjust themselves by using technology to increase their competitiveness. It should set up a learning center for digital technology usage, especially for the entrepreneurs, or coordinate with network partners to provide training on digital technology and online commerce for them. There should also be free Wi-Fi provision around CBT attractions.

Secondly, finance and investment: due to the liquidity problem, the entrepreneurs cannot invest more, and they are also blacklisted because of a nonperforming loan (NPL) that prevents them from accessing financial resources. In addition, they do not have enough capacity to invest in high-cost digital technology. Therefore, the government should increase their access to financial sources through government mechanisms and agencies with related missions, while seeking to improve digital technology infrastructure, decreasing service charges, and formulating the policy of tax deduction in some items for SMEs in the service sector.

Thirdly, economy, society, and environment: there are now few tourists or customers due to the economic depression, the COVID-19 pandemic, and natural disasters, which have directly impacted the income of SMEs. Therefore, the government should focus on implementing a policy to revitalize the national economy by promoting all forms of marketing for tourism within the country.

Finally, the support system and mechanisms from the government: due to delayed and discontinued support from the government, and the restrictions of the budget regulations of related government agencies which prevent the integration of the budgets and missions, SMEs' promotion and development by the government sector are delayed and they cannot cope with the rapidly changing business conditions. Moreover, the access system to the government database is still complex and some data are unclear, as they cannot be analyzed in order for a marketing plan to be set. Therefore, the government should not only have a clear and systematic SME promotion and development policy, but should also reduce the work process to provide faster support for handling the situation in time. The government

should also focus on developing the skills of government officials in SME promotion and development through digital technology.

*5.6. Future Direction of Research Studies*

Due to the limitations of this study, further studies should focus on the following. Firstly, research should be conducted on the size of change in the external environment during the age of the digital economy in order to analyze the trend of responses by SMEs to the magnitude or changed level of the digital economy's impact. Further studies can also focus on different dimensions of the impact and select the target group of SMEs in the production sector and the trading sector. Secondly, studies should be conducted on the participation level of the stakeholders in CBT development and promotion and on measuring the happiness of both the community and tourists as a result of community tourism management. One potential topic is examining how to formulate happiness indicators quantitatively and qualitatively in order to reflect the impacts and outcomes from CBT policy implementation. Finally, further research can study the gap in the service quality of businesses in the service sector of CBT by focusing on the impacts and outcomes after the improvement of service quality in order to help entrepreneurs, especially from community enterprises, to seek marketing strategies which respond to customers' needs.

**Author Contributions:** Conceptualization, W.T.; Data curation, W.T.; Formal analysis, W.T.; Investigation, W.T.; Methodology, W.T.; Project administration, W.T.; Supervision, C.-n.M.; Writing—original draft, W.T.; Writing—review and editing, W.T. All authors have read and agreed to the published version of the manuscript.

**Funding:** This research was funded by National Institute of Development Administration (NIDA) under the NIDA regulation on scholarships and awards for education B.E. 2550 and the resolution of the sub-committees of institutional administration (3/2020) on 11 March 2020.

**Institutional Review Board Statement:** The study was approved by the Ethics Committee in Human Research, National Institute of Development Administration on 28 April 2020 (certificate of approval number 2020/0023).

**Informed Consent Statement:** Informed consent was obtained from all subjects involved in the study.

**Data Availability Statement:** The study did not report any data.

**Conflicts of Interest:** The authors declare no conflict of interest. The funders had no role in the design of the study; in the collection, analyses, or interpretation of data; in the writing of the manuscript, or in the decision to publish the results.

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
