# Peer review of "Thai SMEs’ Response in the Digital Economy Age: A Case Study of Community-Based Tourism Policy Implementation"

_socsci, doi:10.3390/socsci11040180_

Round 1

Reviewer 1 Report

The article is interesting and a good candidate for publication. Although the authors tried to integrate a lot of topics, resulting in a confusing document and with poor theoretical support. 

I recognize the efforts done in the first revision (which I didn't review). As such, I consider that two more topics should be addressed by the authors:

  1. I think there is excessive weight or focus on the Thai industry. One thing is the research field, another is the global relevance of the paper. If the authors are contributing to the academic knowledge, they should abstract in order to prepare a document useful to other academics and practitioners around the globe. I suggest a revision of the document in this line. For example, the authors say that "The theoretical contributions of the factors which together influenced the Thai SMEs’ response in the digital economy age can be summarized as follows. " According to my previous argument, this does not make sense.

2. The implications at the end of the conclusions are very weak, and should be extended to provide concise advises for decision-makers.

Author Response

Response to Reviewer 1 Comments

Point 1: Is the content succinctly described and contextualized with respect to previous and present theoretical background and empirical research (if applicable) on the topic?

 Response 1: I revised the content succinctly more described and contextualized with respect to previous and present theoretical background and empirical research on the topic “2. Literature Review”. [line: 55-157 and in the Table 1.]

Point 2: Are the arguments and discussion of findings coherent, balanced and compelling?

Response 2: I already revised the arguments and discussion of findings to be more coherent, balanced, compelling, and clear. [line 254-325]

Point 3: Are the conclusions thoroughly supported by the results presented in the article or referenced in secondary literature?

Response 3: I revised the conclusions thoroughly supported by the results presented in the article or referenced in the secondary literature on the topic “5. Conclusion and Recommendations”. [451-584]

Point 4: I think there is excessive weight or focus on the Thai industry. One thing is the research field, another is the global relevance of the paper. If the authors are contributing to the academic knowledge, they should abstract in order to prepare a document useful to other academics and practitioners around the globe. I suggest a revision of the document in this line. For example, the authors say that "The theoretical contributions of the factors which together influenced the Thai SMEs’ response in the digital economy age can be summarized as follows. " According to my previous argument, this does not make sense.

Response 4: I avoided using the word “Thai” by using only theses words “SMEs’ response” when I have to explain theoretical contributions of the factors which together influenced the SMEs’ response in the digital economy age on the topic “4. Findings and Discussion” and “5. Conclusions and Recommendations”

Point 5: The implications at the end of the conclusions are very weak, and should be extended to provide concise advises for decision-makers.

Response 5: I revised the end of the conclusions. In accordance with the results of the study of problems, obstacles, and suggestions from the stakeholders for the promotion and development of SMEs in the service sector of CBT, the policy recommendations were made in four dimensions. [line 551-584]

Reviewer 2 Report

Literature review
I suggest to merge the section “Literature review” with the“Conceptual framework”  included in the research methods [line: 167-190].

Research method
It would be good if the author could state the total amount of CBT existing businesses in the case studies and what percentage in relation to such a total is represented by the “At least 40 entrepreneurs from each community for the survey” [Table: 2].

It would be good, also, to mention the total number of tourists that visit the studied areas and what percentage in relation to such a total is represented by the “100 representatives of tourists from each
community for the survey” [Table: 2].

Results
Please include more results from the survey about the “100 representatives of tourists from each community” [Lines: 414-418].

Conclusions and recommendations
Expand and discuss more around the “16.4. The Gap of Service Quality of the CBT Business” [line: 459].

Others: supplementary material
Due to the international public of the journal, I suggest to include a map with the geographical situation of the businesses that are mentioned or analyzed in the paper.

Author Response

Response to Reviewer 2 Comments

Point 1: Is the content succinctly described and contextualized with respect to previous and present theoretical background and empirical research (if applicable) on the topic?

Response 1: I revised the content succinctly more described and contextualized with respect to previous and present theoretical background and empirical research on the topic “2. Literature Review”. [line: 55-157 and in the Table 1. Independent variables in the study.]

Point 2: Are the research design, questions, hypotheses and methods clearly stated?

Response 2: I revised the content on the topic “3. Research Method” [line 144-208], especially writing the hypotheses of the study to be clear. [line 182-191]

Point 3: Are the arguments and discussion of findings coherent, balanced and compelling? For empirical research, are the results clearly presented?

Response 3: I already revised the arguments and discussion of findings to be more coherent, balanced, compelling, and clear. [line 254-325]

Point 4: Are the conclusions thoroughly supported by the results presented in the article or referenced in secondary literature?

Response 4: I revised the conclusions thoroughly supported by the results presented in the article or referenced in the secondary literature on the topic “5. Conclusion and Recommendations”. [451-550]

Point 5: Literature review. I suggest to merge the section “Literature review” with the “Conceptual framework” included in the research methods [line: 167-190].

Response 5: I already merged the section “Literature Review” with the “Conceptual framework” included in the research methods.

Point 6: Research method. It would be good if the author could state the total amount of CBT existing businesses in the case studies and what percentage in relation to such a total is represented by the “At least 40 entrepreneurs from each community for the survey” [Table: 2].

Response 6: I already stated approximately CBT existing businesses in the case studies and the percentage in relation to such a total in Table 2.

Point 7: It would be good, also, to mention the total number of tourists that visit the studied areas and what percentage in relation to such a total is represented by the “100 representatives of tourists from each community for the survey” [Table: 2].

Response 7: I would like to explain more that I could not reach the total number of tourists because it is an infinite population. I, therefore, stated this condition under Table 2.

Point 8: Results. Please include more results from the survey about the “100 representatives of tourists from each community” [Lines: 414-418].

Response 8: I included more results and already stated: “from 400 representatives from four communities”. [line 439-450]

Point 9: Conclusions and recommendations. Expand and discuss more around the “16.4. The Gap of Service Quality of the CBT Business” [line: 459].

Response 9: I already revised the content of the topic “5.4 . The Gap of Service Quality of the CBT Business”. [line 540-550]

Point 10: Others: supplementary material. Due to the international public of the journal, I suggest to include a map with the geographical situation of the businesses that are mentioned or analyzed in the paper.

Response 10: I already prepared a map with the geographical situation of the businesses that are mentioned in the paper.
